# Continence Status and Presence of Pressure Skin Injury among Special Elderly Nursing Home Residents in Japan: A Nationwide Cross-Sectional Survey

**DOI:** 10.3390/geriatrics6020034

**Published:** 2021-03-26

**Authors:** Motofumi Suzuki, Megumi Kodaira, Keiko Suyama, Taro Murata, Haruki Kume

**Affiliations:** 1Department of Urology, Tokyo Metropolitan Bokutoh Hospital, Tokyo 130-8575, Japan; 2Graduate School of Health and Welfare Sciences, International University of Health and Welfare, Tokyo 107-0052, Japan; kodaira@iuhw.ac.jp; 3Graduate School of Medicine Program for Nursing and Health Sciences, Ehime University Graduate School of Medicine, Ehime 790-8577, Japan; kankeiko@m.ehime-u.ac.jp; 4Department of Urology, Tokyo Teishin Hospital, Tokyo 102-8798, Japan; taromurata1002@gmail.com; 5Department of Urology, Graduate School of Medicine, The University of Tokyo, Tokyo 113-8655, Japan; KUMEH-URO@h.u-tokyo.ac.jp

**Keywords:** urinary incontinence, fecal incontinence, double incontinence, pressure skin injury, special elderly nursing home

## Abstract

Urinary and fecal incontinence as well as skin pressure injury are common healthcare problems in nursing homes; however, the prevalence and related risk factors were not well understood in the Japanese special elderly nursing home settings. We surveyed the prevalence of urinary, fecal and double incontinence, and skin pressure injury among the elderly living in special elderly nursing homes in Japan. A nationwide cross-sectional epidemiological survey was conducted with a total of 4881 residents. The prevalence of urinary, fecal and double incontinence was 82.9%, 68.9% and 64.9%, respectively. Skin pressure injury was found in 283 residents (283/4881, 5.8%). Age, Care-Needs level, loss of voiding desire, and fecal incontinence were significant risk factors for urinary incontinence. Residential period, Care-Needs level, loss of voiding and defecation desires, and urinary incontinence were significant risk factors for fecal incontinence. Only male sex was a significant risk factor for skin pressure injury. Our study revealed continence status and the prevalence of pressure skin injury among older adult residents who receive end-of-life care in special elderly nursing homes in Japan. Further studies should be conducted to examine whether recovery of urinary and fecal sensations improves continence status.

## 1. Introduction

Mean life expectancies of Japanese males and females were 81.41 and 87.45 years in 2019 [1]. In 2019, the proportion of the elderly aged ≥65 years became 28.1% in Japan and the Japanese situation has been in super-aged society (the proportion of the elderly ≥65 years old, >21%) since 2007 [2]. Nowadays, number of the elderly people who need long-term care is approximately 6.7 million (almost equal to 5% of the Japanese population) in Japan and it has also kept increasing [3].

Geriatric syndromes have been defined as multifactorial health conditions that occur when the accumulated effects of impairments in multiple systems render an older person vulnerable to situational challenges [4]. Incontinence is one of the aspects of geriatric syndromes that can be often observed among the elderly individuals living in nursing homes. As for urinary incontinence, not only age-related changes in the lower urinary tract and external genitalia but also other factors outside the lower urinary tract including comorbid medical illness, neurological and psychiatric conditions, some medications, functional impairments, and environmental factors are known risk factors of urinary incontinence. Fecal incontinence usually coexists with urinary incontinence in frail older people. Aside from age, the following are primary risk factors of fecal incontinence: stool consistency, bowel-related disorders, impaired mobility, functional impairment, dementia, neurological diseases, diabetes mellitus, chronic medical conditions, and depression [5]. In the Medicare- or Medicaid-certified nursing homes setting in the United States (US) (*n* = 2,936,146), Ahn et al. reported the following conditions were risk factors for pressure skin injury: younger age, male sex, Black and Hispanic ethnicities, lower body mass index (BMI), anemia, malnutrition, dehydration, urinary incontinence, fecal incontinence, comatose, heart failure, respiratory failure, end-stage renal disease, diabetes mellitus, multiple sclerosis, coronary artery disease, chronic obstructive pulmonary disease, vascular disease, cirrhosis, septicemia, pneumonia, urinary tract infection, multidrug-resistant organism, and lower mobility [6].

There are two types of public long-term care facilities in Japan; geriatric health services facilities and special elderly nursing homes. The former is a facility which provides medical care, nursing care, daily life support, and rehabilitation to help the elderly to go back home. The latter is a facility that provides end-of-life care to support the peaceful and heartwarming lives of the elderly. Recently, we conducted a nationwide survey of continence status among the elderly living in geriatric health services facilities in Japan, and reported that the prevalence rates of urinary, fecal and double incontinence were 66.9%, 42.8% and 41.1%, respectively. Motor and cognitive subscales of the Functional Independence Measure score, an indicator of daily activity of living, was significantly worse among the elderly with incontinence than those without [7]. As far as we know, a nationwide survey regarding continence status among elderly living in Japanese special elderly nursing homes has not been conducted to date. Correlation between loss of voiding and defecation desires and continence status has been scarcely investigated, either. Pressure skin injury is also a common health problem particularly among the physically limited or bedridden elderly [8]. In nursing home settings, the national average prevalence of pressure skin ulcers in the US is reported to be 7.3% [9]. A Japanese hospital-based study reported the prevalence of pressure skin injury as 4.3% [10]; however, the prevalence among the elderly living in Japanese special elderly nursing homes has been still unknown.

The Long-Term Care Insurance System is the public social service for the frail elderly persons who need long-term care in Japan. The Japanese Government started this system in 2000. The Long-Term Care Insurance covers cost of care partially according to the Care-Needs level of each elderly person. The classification process of Care-Needs level starts with an initial assessment by a trained local government assessor, who evaluates his/her care-needs by a questionnaire on current physical and psychological status (73 items) and use of medical procedures (12 items). The results are transformed into the applicant’s standardized scores for the seven dimensions of physical and psychological status. In addition, required time for the nine categories of care (grooming, bathing, eating, toileting, transferring, assistance with instrumental activities of daily living, behavioral problems, rehabilitation, and medical services) is estimated. The Care-Needs level is finally determined by the Nursing Care Needs Certification Board, considering the results of assessment and the statement by the applicant’s primary care physician. The Care-Needs level of this insurance system is a five-point ordinal scale that ranges from level-1 (estimated total care minutes per day, 30–50 min) to level-5 (estimated total care minutes per day, ≥110 min) [11]. To live in a special elderly nursing home, elderly individuals usually require the Care-Needs level-3 (estimated total care minutes per day, 70–90 min) or more. Elderly individual with severe dementia or psychological disorders also can live in a special elderly nursing home even if they are Care-Needs level-1 or -2.

The aim of this study was to measure the prevalence rates of urinary, fecal and double incontinence, as well as pressure skin injury among the elderly living in Japanese special elderly nursing homes. We also investigated voiding and defecation desires of the participants and identified the significant risk factors of continence status and pressure skin injury by including these sensations as confounding factors.

## 2. Materials and Methods

### 2.1. Design

A cross-sectional, nationwide point prevalence measurement study.

### 2.2. Sample and Setting

From 15 October 2015 to 16 November 2015, we recruited elderly residents aged ≥65 years from the member special elderly nursing homes of the Japanese Council of Senior Citizens Welfare Service. All the elderly participants received end-of-life care in each facility.

### 2.3. Data Collection

Caregivers were the main person to provide the data of the elderly participants in the present study. Each member facility randomly selected one-tenth of the total number of residents. All residents aged ≥65 years in each facility were numbered consecutively, and the chief caregiver of each facility asked the residents who were numbered in multiples of 10 to participate in the present study. If the resident refused to participate, the chief caregiver asked the next numbered resident to participate. We did not count the number of residents who refused to participate. We included all residents who agreed with this study without setting any exclusion criteria. Caregivers of each facility collected data by using a structured investigation sheet including sex, age, residential period, continence status, voiding and defecation desires, existence of pressure skin injury, and stool form. The point in time of the outcome variables for urinary incontinence, fecal incontinence, double incontinence and pressure skin injury was on the day of data collection. Urinary incontinence as well as fecal incontinence was defined as any leakage of urine/feces [7]. We assessed the frequency of urinary incontinence and fecal incontinence according to five levels, which are as follows: none, less than once a week, twice or more per week, almost every day, and every day. We modified the classification of domain-1 of the International Consultation on Incontinence Questionnaire-Short Form [12] to create these five levels of incontinence frequency. We also originally assessed voiding and defecation desires by following three levels: normal, ambiguous, and impaired. Residents with normal voiding or defecation desire could go to the bathroom by oneself or ask caregiver for toileting assistance on the right timing. Residents with ambiguous/impaired voiding or defecation desire sometimes/always failed to go to the bathroom or could not ask a caregiver for toileting assistance on the right timing. We also investigated the implementation of toileting assistance for urination. Stool form was assessed by the Bristol stool scale that classifies a stool form into seven categories: type 1, separate hard lumps; type 2, lumpy and sausage like; type 3, a sausage shape with cracks in the surface; type 4, like a smooth, soft sausage or snake; type 5, soft blobs with clear-cut edges; type 6, mushy consistency with ragged edges; and type 7, liquid consistency with no solid pieces [13]. In addition, we simply assessed pressure skin injury as normal, intact skin with nonblanchable redness, and skin ulcers to report the prevalence of pressure skin injuries among the residents living in the special elderly nursing homes in Japan. Caregivers could distinguish intact skin with nonblanchable redness and skin ulcers from intact skin; however, they did not have enough skill to classify the stage of pressure skin injury due to lack of education concerning pressure skin injuries.

### 2.4. Data Analysis

Statistical analysis was performed using the JMP PRO software, version 15.0.0 (SAS, Cary, NC, USA). We evaluated caregiver-reported outcomes of the elderly residents including the prevalence of urinary, fecal and double incontinence, and skin pressure injury among the elderly living in special elderly nursing homes in Japan. Odds ratios and 95% confidence intervals were assessed to identify significant risk factors of urinary incontinence, fecal incontinence, and pressure skin injury via a logistic regression analysis. In multivariate analyses, we included following parameters in the parentheses to detect significant risk factors of urinary incontinence (age, sex, residential period, Care-Needs level, voiding desire, defecation desire, fecal incontinence, and pressure skin injury), fecal incontinence (age, sex, residential period, Care-Needs level, voiding desire, defecation desire, urinary incontinence, and pressure skin injury), double incontinence (age, sex, residential period, Care-Needs level, voiding desire, defecation desire, and pressure skin injury), and pressure skin injury (age, sex, residential period, Care-Needs level, voiding desire, defecation desire, urinary incontinence, and fecal incontinence). A *p* value of <0.05 was considered statistically significant.

### 2.5. Ethical Considerations

This study was approved by the ethics committee of Tokyo Teishin Hospital (approval No. 975). A written informed consent was obtained from all participants or their family members.

## 3. Results

### 3.1. Characteristics of the Elderly Participants

We asked 2341-member special elderly nursing homes registered in the Japanese Council of Senior Citizens Welfare Service to be involved in the present study, and 969 (41.4%) facilities responded. In total, 5673 elderly residents from 969 facilities participated in the study. However, 792 residents were excluded from the analysis based on the following exclusion criteria: aged 64 years or younger (*n* = 54); missing information of age (*n* = 23), sex (*n* = 8), residential period (*n* = 53), Care-Needs level (*n* = 94), urinary incontinence (*n* = 145), fecal incontinence (*n* = 119), pressure skin injury (*n* = 175), voiding desire (*n* = 55), and defecation desire (*n* = 66). The characteristics of the study participants are shown in Table 1.

Number of elderly individuals with normal voiding desire and defecation desire was limited to 1217 (24.9%). In other words, three quarters of participants had impaired voiding and/or defecation desire. The prevalence rates of urinary, fecal and double incontinence were 82.9% (4045/4881 participants), 68.9% (3364/4881 participants), and 64.9% (3169/4881 participants), respectively. Only 641 (13.1%) participants had continence, while 4240 (86.9%) suffered from urinary and/or fecal incontinence. After excluding 30 participants with missing data of toileting assistance from 4881 participants (*n* = 4851), toileting assistance was conducted in 2144 (44.2%) and not conducted in 2707 (55.8%) elderly individuals. Of 2144 participants with toileting assistance, types of toileting assistance conducted was timed voiding in 1271 (59.3%), prompted voiding in 571 (26.6%), and unknown in 302 (14.1%), respectively. Of 2707 participants without toileting assistance, 660 (24.4%) could go to the bathroom independently, 1222 (45.1%) usually urinate in diapers or pads, 60 (2.2%) refused toileting assistance, but the reasons for 765 (28.3%) subjects were unknown.

Overall prevalence rate of pressure skin injury was 5.8% (283/4881 participants). The number of participants with intact skin with nonblanchable redness and skin ulcers by body sites were as follows: 4 and 0 in head, 1 and 1 in shoulder, 5 and 2 in elbow, 10 and 5 in back, 123 and 99 in hip/sacrum, and 1 and 1 in lower limb, respectively. Hip/sacrum was the most frequent site of pressure skin injury (5.8%, 283/4881 participants) with intact skin with nonblanchable redness (2.5%, 123/4881 participants) and skin ulcers (2.0%, 99/4881 participants). As for the shape of the stool, the proportion of elderly with normal shape of the stool (Bristol stool scale types 3 and 4) was 38.6% (1884/4881 participants).

### 3.2. Risk Factors of Urinary Incontinence, Fecal Incontinence, Double Incontinence, and Pressure Skin Injury

In a multivariate analysis, age, Care-Needs level, loss of voiding desire, and fecal incontinence were significantly associated with urinary incontinence (Table 2). Meanwhile, residential period, loss of voiding desire, loss of defecation desire, and urinary incontinence were significant risk factors for fecal incontinence (Table 3). Moreover, age, residential period, loss of voiding desire, and loss of defecation desire were significant risk factors for double incontinence (Table 4). Concerning with pressure skin injury, only male sex was a significant risk factor (Table 5).

## 4. Discussion

This is the first nationwide epidemiological survey of continence status and pressure skin injury as well as loss of voiding/defecation desire among the elderly living in special elderly nursing homes in Japan. We also investigated types of continence care provided by caregivers. The prevalence rate of urinary, fecal and double incontinence was 82.9%, 68.9% and 64.9%, respectively. In our previous study conducted in geriatric health services facilities in Japan, the prevalence of urinary, fecal and double incontinence was 66.9%, 42.8% and 41.1%, respectively [7]. Continence status of the present study was worse than the previous study. It might be because of older age (mean age; 87.0 versus 85.2 years) and higher dependency of care (mean Care-Needs level (continuous); 3.9 versus 2.9) of the present study. Offermans et al. reviewed 10 epidemiological studies conducted in the US, Germany, Italy, Switzerland, France, Sweden, and Japan that addressed the prevalence of urinary incontinence in nursing home settings. They reported that the prevalence rate of urinary incontinence ranged from to 47% to 77% and identified age, female sex, and limited cognitive function were associated risk factors for urinary incontinence [14]. In the present study, age, female sex, the Care-Needs level, loss of voiding desire and fecal incontinence were significantly associated with urinary incontinence. Saga et al. reported the prevalence of not only urinary incontinence but also fecal incontinence and double incontinence among the 898 residents with mean age of 85.5 years living in Norwegian nursing homes. The prevalence rate of urinary, fecal and double incontinence was 72.0%, 42.8%, and 40.2%, respectively. Continent residents were characterized by being in short-term care, shorter stay in nursing home, less cognitive and physical impairment, less Parkinson’s disease, stroke, constipation and diarrhea, and more independence in activities of daily living [15]. In our study participants, continent residents were characterized by being of younger age, lower care dependency, intact voiding, and defecation desires (data not shown). Higami et al. reported the prevalence of urinary, fecal and double incontinence among the elderly cognitively impaired older residents in long-term care facilities in East Asia including Japan, South-Korea, China, Taiwan, and Thailand. The prevalence rate of urinary, fecal and double incontinence ranged from 10.1% (Taiwan) to 71.0% (South-Korea), 4.0% (Taiwan) to 57.0% (South-Korea) and 4.0% (Taiwan) to 57.0% (South-Korea), respectively [16]. Gorina et al. reported that the percentage of difficulty controlling urinary bladder and/or bowels among the US hospice patients aged 65 years and over was 82.4% [17]. Similarly, 86.9% (4240/4881 participants) of our study participants suffered from urinary and/or fecal incontinence. Both urinary and fecal incontinence are highly prevalent conditions among the elderly in the stage of end-of-life. The prevalence rate of urinary, fecal and double incontinence in our study participants marked the highest rate compared to the previous reports. Worsened Care-Needs level and loss of voiding desire were significant common risk factors for urinary, fecal and double incontinence of our study participants.

Prompted voiding is a toileting assistance that was recommended by the International Consultation on Incontinence as an effective intervention for the continence care of nursing home residents and home-care clients [18]. In prompted voiding care, caregivers regularly ask the elderly individual whether he/she feels voiding desire or not. If the response is yes, caregiver guides the elderly person to void and provides positive feedback for appropriate voiding. It results in increasing self-initiated toileting and decreased incontinent episodes in the short-term [19]. Meanwhile, timed voiding is a toileting assistance guiding the elderly to the toilet on a fixed time-interval regardless of voiding desire. Timed voiding was the most frequently adopted toileting assistance in the present study followed by prompted voiding and approximately one-quarter of the participants were applied with absorbent products only. We wonder if we manage the elderly with incontinence solely by using absorbent products, they could not recover voiding desire or would lose urinary sensation. From our limited experience, old incontinent elderly individuals recovered urinary sensation after initiating prompted voiding care and then consumption of absorbent products decreased. As suggested by Borglin et al., many nurses understand the importance of continence care in the elderly receiving home-care, either in their own home or in an assisted living facility; however, provision of absorbent products prescription is being the first line of action but not evidence-based interventions, such as timed voiding or prompted voiding. We deeply agree with their following conclusion: “the provision of continence care that is based on key nursing standards, such as evidence-based and person-centered care, as well as individualized continence care that is based on evidenced-based guidelines, would ensure an improvement in the continence care that is presently on offer to older people” [20]. We hope such an approach will evoke and regain urinary or fecal sensation of the elderly. It is useful to try if recovery of urinary and fecal sensations improves continence status and quality of life of the elderly.

In the present study, pressure skin injury was observed in 283 residents (283/4881, 5.8%) and only male sex was a significant risk factor for pressure skin injury. The prevalence was less frequent than the US situation (5.8% versus 7.3%) [9]. In the review article concerning assessment and management of pressure skin ulcers in the elderly, Jaul reported that aging, comorbidities, functional impairments (immobility, incontinence and impaired cognition), malnutrition, and social, family and emotional factors were risk factors of pressure ulcers [8]. These risk factors except for age and continence status were not included in our dataset, therefore, we could not clarify confounding factors developing pressure skin injuries. However, urinary incontinence or fecal incontinence themselves were not a risk factor of pressure skin injury in our study participants.

The present study had several limitations which included lack of BMI, laboratory data which reflects nutritional conditions (blood count, total protein, serum albumin, choline esterase, total cholesterol, and so on), comorbidities (presence of cerebrovascular disease, spinal disease, musculoskeletal disorders, and so on), medications, physical and cognitive functions, and voiding/defecation desires evaluated by global indicators. Multiple chronic diseases and complicating factors (morbidity and fragility) are associated with malnutrition and pressure skin injury. Moreover, severity of pressure skin injury was not assessed by a validated manner. We need to consider the etiology of pressure skin injuries in future studies. Sampling process was another limitation of the present study. Complete enumeration survey or sampling using random number tables was necessary to reduce a sampling bias. Nevertheless, participation of nationwide large number of elderly persons with information of voiding and defecation desires was a strength of the present study. Further studies should be conducted to examine whether recovery of urinary and fecal sensations improves continence status.

## 5. Conclusions

In conclusion, the present study revealed continence status and the prevalence of pressure skin injuries among older adult residents who receive end-of-life care in special elderly nursing homes in Japan.

## Figures and Tables

**Table 1 geriatrics-06-00034-t001:** Baseline characteristics of the elderly participants.

Characteristics		Total Number of Elderly Participants *n* = 4881
Age, Years (mean ± SD)		87.0 ± 7.5
Sex, *n* (%)	Male/Female	934 (19.1%)/3947 (80.9%)
Residential Period, Years (Median [IQR])		2.6 [1.2 to 4.8]
Care-Needs level, *n* (%)	Level-1 Level-2 Level-3 Level-4 Level-5	112 (2.3%) 323 (6.6%) 1106 (22.7%) 1711 (35.1%) 1629 (33.4%)
Voiding desire, *n* (%)	Normal Ambiguous No complaint of voiding	1356 (27.8%) 1478 (30.3%) 2047 (41.9%)
Defecation desire, *n* (%)	Normal Ambiguous No complaint of voiding	1559 (31.9%) 1409 (28.9%) 1913 (39.2%)
Urinary incontinence, *n* (%)	None Once a week Twice or more per week Almost every day Every day	836 (17.1%) 409 (8.4%) 566 (11.6%) 878 (18.0%) 2192 (44.9%)
Fecal incontinence, *n* (%)	None Once a week Twice or more per week Almost every day Every day	1517 (31.1%) 1048 (21.5%) 1174 (24.1%) 300 (6.1%) 842 (17.3%)
Pressure skin injury, *n* (%)	No Yes Redness Skin ulcer Both Unclassified	4598 (94.2%) 283 (5.8%) 143 (2.9%) 128 (2.6%) 14 (0.3%) 26 (0.5%)
Bristol stool scale, *n* (%)	Type 1 Type 2 Type 3 Type 4 Type 5 Type 6 Type 7 Unclassified	68 (1.4%) 130 (2.7%) 293 (6.1%) 1591 (33.1%) 1927 (40.1%) 626 (13.0%) 123 (2.6%) 42 (0.9%)

Abbreviations: SD, standard deviation.

**Table 2 geriatrics-06-00034-t002:** Risk factors of urinary incontinence.

Variables	Univariate OR (95% CI)	Multivariate OR (95% CI)
Age	1.030 (1.020–1.040)	1.023 (1.011–1.036)
Sex Female Male	Reference 0.760 (0.635–0.910)	Reference 0.842 (0.672–1.053)
Residential period, years	1.039 (1.014–1.064)	1.000 (0.975–1.027)
Care-Needs level (continuous)	4.183 (3.505–4.993)	1.149 (1.041–1.267)
Voiding desire Normal Ambiguous No complaint of voiding	Reference 13.929 (10.840–17.898) 9.277 (7.643–11.260)	Reference 7.256 (5.286–9.987) 2.481 (1.643–3.748)
Defecation desire Normal Ambiguous No complaint of defecation	Reference 7.301 (5.856–9.104) 8.063 (6.584–9.874)	Reference 0.916 (0.664–1.264) 1.159 (0.745–1.804)
Fecal incontinence No Yes	Reference 11.892 (9.964–14.193)	Reference 6.381 (5.158–7.894)
Pressure skin injury No Yes	Reference 2.114 (1.402–3.186)	Reference 1.054 (0.665–1.673)

Abbreviations: OR, odds ratio; CI, confidence interval.

**Table 3 geriatrics-06-00034-t003:** Risk factors of fecal incontinence.

Variables	Univariate OR (95% CI)	Multivariate OR (95% CI)
Age	1.016 (1.007–1.024)	1.000 (0.990–1.011)
Sex Female Male	Reference 0.881 (0.757–1.026)	Reference 1.095 (0.894–1.342)
Residential period, years	1.072 (1.051–1.092)	1.033 (1.009–1.057)
Care-Needs level (continuous)	2.243 (2.097–2.400)	1.143 (1.044–1.252)
Voiding desire Normal Ambiguous No complaint of voiding	Reference 7.304 (6.184–8.627) 18.770 (15.662–22.496)	Reference 1.408 (1.119–1.772) 2.224 (1.589–3.113)
Defecation desire Normal Ambiguous No complaint of defecation	Reference 10.480 (8.804–12.476) 19.740 (16.435–23.708)	Reference 5.339 (4.246–6.713) 6.254 (4.480–8.731)
Urinary incontinence No Yes	Reference 11.892 (9.964–14.193)	Reference 6.386 (5.169–7.890)
Pressure skin injury No Yes	Reference 2.633 (1.892–3.665)	Reference 1.368 (0.927–2.020)

Abbreviations: OR, odds ratio; CI, confidence interval.

**Table 4 geriatrics-06-00034-t004:** Risk factors of double incontinence.

Variables	Univariate OR (95% CI)	Multivariate OR (95% CI)
Age	1.020 (1.012–1.028)	1.011 (1.001–1.022)
Sex Female Male	Reference 0.821 (0.709–0.952)	Reference 0.961 (0.798–1.158)
Residential period, years	1.070 (1.051–1.090)	1.035 (1.013–1.057)
Care-Needs level (continuous)	2.188 (2.048–2.337)	1.188 (1.090–1.294)
Voiding desire Normal Ambiguous No complaint of voiding	Reference 8.827 (7.449–10.461) 16.376 (13.794–19.441)	Reference 2.960 (2.384–3.675) 2.420 (1.786–3.279)
Defecation desire Normal Ambiguous No complaint of defecation	Reference 10.325 (8.710–12.241) 17.447 (14.690–20.721)	Reference 4.712 (3.805–5.835) 7.184 (5.299–9.739)
Pressure skin injury No Yes	Reference 2.230 (1.657–3.000)	Reference 1.191 (0.851–1.667)

Abbreviations: OR, odds ratio; CI, confidence interval.

**Table 5 geriatrics-06-00034-t005:** Risk factors of pressure skin injury.

Variables	Univariate OR (95% CI)	Multivariate OR (95% CI)
Age	1.011 (0.995–1.027)	1.012 (0.995–1.029)
Sex Female Male	Reference 1.335 (1.005–1.773)	Reference 1.550 (1.148–2.092)
Residential period, years	1.030 (1.001–1.061)	1.010 (0.979–1.042)
Care-Needs level (continuous)	1.570 (1.363–1.809)	1.180 (0.994–1.401)
Voiding desire Normal Ambiguous No complaint of voiding	Reference 2.678 (1.729–4.148) 4.461 (2.976–6.689)	Reference 1.397 (0.776–2.516) 1.707 (0.880–3.312)
Defecation desire Normal Ambiguous No complaint of defecation	Reference 2.808 (1.871–4.213) 4.262 (2.928–6.203)	Reference 1.652 (0.952–2.867) 1.876 (0.996–3.533)
Urinary incontinence No Yes	Reference 2.114 (1.402–3.186)	Reference 1.056 (0.666–1.673)
Fecal incontinence No Yes	Reference 2.633 (1.892–3.665)	Reference 1.372 (0.927–2.031)

Abbreviations: OR, odds ratio; CI, confidence interval.

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
