# Peer review of "Continence Status and Presence of Pressure Skin Injury among Special Elderly Nursing Home Residents in Japan: A Nationwide Cross-Sectional Survey"

_geriatrics, 2021, doi:10.3390/geriatrics6020034_

Round 1

Reviewer 1 Report

Here they are my comments.

To make the methods section more readable, it is suggested to divide it using subheadings as follows and bring the related information under each: Design, Sample and setting, Data collection, Data analysis, Ethical considerations.

The inclusion and exclusion criteria should be stated.

Line 71, please give details on random selection. 

Lines 71-76, how the data collection tool has been developed, and how its validity and relaibility has been assessed?

Line 85, the levels: normal, ambiguous, and impaired should be defined practically.

Line 86, again pressure skin injury should be defined and stated how practically they have been examined and measured. How many persons have been involved in the measurement and if inter-rater reliability has been assessed before the measurment?

Lines 88-103, I could not understand their linkes to your study.

In the results, please specify the response rate for participation in this research.

Tables are disorganised and should be justered to able to read and interprete scores.

In the discussion, I can ses that you have used 'our cohort'. What is it?

Throughout the discussion, when you cite a similar study for the comparison of your findings, please decsribe the details of samples, setting and country. Perhaps a thorough literature would help you improve the variation in studies used here for citation and comparison. 

Conclusion should be improved. What would be the practical implications of your research for practice, education, policy making, and management?

Suggestions for future studies are needed. 

Author Response

Thank you for your critical comments. Please see the attachment.

Reviewer 2 Report

I would like to commend researchers for their interest in knowing the prevalence of incontinence and skin ulcers in elderly nursing home settings. The manuscript is understandable in all the sections, the results correspond to the objectives set.

Comments:

The introduction is brief and in particular the issue of pressure ulcers is not addressed in depth, for instance the etiology of pressure ulcers in the elderly has not been considered. Multiple chronic diseases and complicating factors (morbidity and fragility) are associated with malnutrition and pressure ulcers.  In this sense, the study has several limitations that are discussed very lightly. We don’t know the cause of admission of participants in the elderly nursing homes. Were the participants at the end of their lives? We don't know the reason why data on the nutritional state were not collected, only it’s commented that the body mass index was not collected. It’s recommended to explain all in the limitations section.

Author Response

(The authors gave the same response as above.)

Round 2

Reviewer 1 Report

Nothing more.

Author Response

thanks